# Determination of Long-Term Soil Apparent Thermal Diffusivity Using Near-Surface Soil Temperature on the Tibetan Plateau

Bing Tong [1], Hui Xu [1], Robert Horton [2], Lingen Bian [3] and Jianping Guo [1,*]

1   State Key Laboratory of Severe Weather, Chinese Academy of Meteorological Sciences, Beijing 100081, China
2   Agronomy Department, Iowa State University, Ames, IA 50011, USA
3   Chinese Academy of Meteorological Sciences, Beijing 100081, China
*   Correspondence: jpguo@cma.gov.cn

**Abstract:** The knowledge of soil apparent thermal diffusivity (k) is important for investigating soil surface heat transfer and temperature. Long-term k determined using the near-surface soil temperature is limited on the Tibetan Plateau (TP). The main objective of this study is to determine k with a conduction–convection method using the near-surface soil temperature measured at three sites during 2014–2016 on the TP. The hourly, daily, and monthly k values of the 0.0 m to 0.20 m layer were obtained. The hourly and daily k values ranged from $0.3 \times 10^{-6}$ m$^2$ s$^{-1}$ to $1.9 \times 10^{-6}$ m$^2$ s$^{-1}$ at the wet site, and from $1.0 \times 10^{-7}$ m$^2$ s$^{-1}$ to $4.0 \times 10^{-7}$ m$^2$ s$^{-1}$ at the two dry sites. For the monthly timescale, k ranged from $0.4\ (\pm 0.0) \times 10^{-6}$ m$^2$ s$^{-1}$ to $1.1\ (\pm 0.2) \times 10^{-6}$ m$^2$ s$^{-1}$ at the wet site, and varied between $1.7\ (\pm 0.0) \times 10^{-7}$ m$^2$ s$^{-1}$ and $3.3\ (\pm 0.2) \times 10^{-7}$ m$^2$ s$^{-1}$ at the two dry sites. The k was not constant over a day, and it varied seasonally to different degrees at different sites and years. The variation of k with soil moisture (θ) appeared to be roughly similar for unfrozen soil at these sites and years, namely, k increased sharply before reaching the peak as θ increased, and then it tended to be stable or varied slightly with further increases in θ. This variation trend was consistent with previous studies. However, the relationship between k and θ changed when soil temperature was below 0 °C, because ice had higher k than water. The correlation coefficients (r) between k and θ ranged from 0.37 to 0.80, and 0.80 to 0.92 on hourly and monthly timescales, respectively. The monthly and annual k values were significantly correlated (r: 0.73~0.93) to the Normalized Difference Vegetation Index (NDVI). The results broaden our understanding of the relationship between in situ k and θ. The presented values of k at various timescales can be used as soil parameters when modeling land–atmosphere interactions at these TP regions.

**Keywords:** soil thermal diffusivity; conduction–convection method; soil temperature; soil water content; Normalized Difference Vegetation Index (NDVI)

## 1. Introduction

Soil apparent thermal diffusivity (k), defined as the ratio of soil thermal conductivity (λ) to volumetric heat capacity (Cv), is the parameter that describes the rate of transmission of temperature change within the soil [1]. It is associated with transient processes of heat conduction and intra-porous convection [2,3]. Knowledge of k is essential for estimating soil temperature, which plays an important role in regulating land surface processes [4–8], estimating soil heat flux [3,9,10], and simulating permafrost extent [11].

Soil apparent thermal diffusivity can be measured in a laboratory with a commercial (e.g., KD2, METER Group, Inc., Pullman, WA, USA) or homemade instrument (e.g., heat pulse probe, [12,13]), or estimated by k empirical models using soil texture, moisture (θ), bulk density, soil organic matter, temperature, vegetation index, etc. (e.g., [14–17]). Obtaining k by sampling and analyzing soil from multiple depths and locations is highly valuable, but it is also time consuming, invasive, difficult to repeat over time, and costly [18]. Heat pulse probe, commonly used in laboratory, has been recognized as reliable tool for

measuring soil thermal properties in the field (e.g., [19,20]). The small sampling volume and relatively sophisticated equipment setup, however, limit its extensive application in field conditions [17]. The k empirical models require various soil specific parameters and some of the parameters are difficult to determine in situ, which limits their use.

Besides, there are numerous studies focused on determining k for time periods (e.g., diurnal, or annual) using near surface soil temperature measured in situ at multiple depths (e.g., [3,21–28]). Conduction is the primary heat transfer mechanism in soil [29]. Some methods are based on solutions of one-dimensional conduction heat transfer equations with constant diffusivity and soil upper boundary described by a sinusoidal function [30], by two harmonics [31,32], or by a Fourier series [21]. Horton et al. [21] examined these k methods and reported that the harmonic method performed best. Since soil is an intra-porous medium, soil heat transfer occurs via a complex combination of conductive and convective heat transfer processes [3]. Considering this, some researchers developed methods of estimating k based on the solution of a one-dimensional soil conduction–convection heat transfer equation with the soil upper boundary described by a sinusoidal function [23,24,33], or by a Fourier series [34]. Wang et al. [35] compared the six k methods (the amplitude, phase, arctangent, logarithmic, conduction–convection, and harmonic) at a site in the Loess Plateau of China, and suggested that the harmonic method performed best, and the conduction–convection method followed. Compared with the harmonic method, the conduction–convection method had a less accurate description of the upper boundary temperature, but it included more physics in the soil heat transfer process. By comparisons of the soil temperature estimation at the 0.10 m depth in the permafrost regions of Qinghai-Xizang (Tibet) Plateau, Hu et al. [36] found the conduction–convection method [23] performed better than the conduction methods (amplitude and phase method). Compared with the k method proposed by Hu et al. [34], the k method proposed by Gao et al. [23] is easier to calculate and has an explicit mathematical expression for k. Some researchers also developed analytical [37] and numerical [38–40] solutions for estimating the k of nonuniform soils. Due to some limitations and complexities, these methods have not been widely used in various fields. The k methods mentioned above with soil temperature can only obtain daily or longer timescale k values. The shorter timescales, e.g., hourly k values, remain unknown.

The Tibet Plateau (TP) is known as the "third pole" of the earth, with an average altitude of over 4000 m [41]. Land–atmosphere interactions over the TP play a crucial role in controlling the regional and global climate [42–45]. The soil thermal properties exert import roles in soil heat and water transfer, which have significant effects on land–atmosphere interactions [46,47]. Wang et al. [48] derived the soil thermal properties of the 0.025–0.075 m layer with soil data measured over two years at a cold semi-desert site on the western TP, and examined their relationships with soil moisture. They calculated k as the ratio of $\lambda/C_v$, which were determined by soil temperature and heat fluxes measured at two depths [49]. Gao [26] determined the k of the 0.015–0.05 m layer with the conduction–convection method at BJ on the TP during day of year (DOY) 195 to 258, 1998. Using the same k method, recently Zhou et al. [28] determined the daily k with soil temperatures measured in the 0.8 m and 3.2 m soil layer at 39 weather sites in the TP during 1980 to 2001 and examined the spatio-temporal variation of k.

Due to the harsh natural conditions and limited observations in the TP, few studies have focused on soil apparent thermal diffusivity determined with long-term in situ near-surface soil data. To fill the gap, the objective of this study is to determine long-term soil apparent thermal diffusivity using the conduction–convection method [23], with the near-surface soil temperature measured at three sites during 2014–2016 over the TP. The specific objectives are twofold, (1) to determine k at different timescales (e.g., hourly, daily, and monthly) over 2014–2016 at the three sites; (2) to examine the relationship between k and soil moisture at different timescales. The conduction–convection method is chosen to determine k because it considers more physics in the soil heat transfer process compared to conduction methods, and it provides an explicit mathematical expression

for k as a function of soil temperature amplitude and phase. To obtain sub-daily k, the Dynamic Harmonic Regression (DHR, [50,51]) is used to extract sub-daily soil temperature parameters (amplitude and phase) from the soil temperature series data. In order to indirectly investigate the effect of the soil moisture on k from the vegetation aspect, the monthly relationship between k and the Normalized Difference Vegetation Index (NDVI) is also investigated.

## 2. Materials and Methods

### 2.1. Site Description

The data used in this study are from three TP sites: the Bijie site of Nagqu Station of Plateau Climate and Environment, Chinese Academy of Sciences (NPCE-BJ, hereinafter abbreviated as BJ) in the central part of the TP, the Qomolangma Atmospheric and Environmental Observation and Research Station, CAS (QOMS) in the south part of the TP, and the Ngari Desert Observation and Research Station, CAS (NADORS) in the northwestern part of the TP. The geographic characteristics of the three sites are listed in Table 1, and the locations of the three sites are shown in Figure 1. The BJ site is in a flat, open prairie except for the north, where a low hill stands. This site is well covered with grass and a canopy height up to 5 cm. The other two sites are barren and the ground is relatively flat and open, with sparse and short vegetation. Additional information about the sites can be found in Ma et al. [52].

**Table 1.** List of the geographic characteristics of the three sites.

| Site | Latitude | Longitude | Elevation (m) | Land Cover | Soil Type |
|------|----------|-----------|---------------|------------|-----------|
| BJ | 31.37°N | 91.90°E | 4509 | Alpine meadow | Sandy silt loam |
| QOMS | 28.36°N | 86.95°E | 4298 | Alpine desert | Sand and gravel |
| NADORS | 33.39°N | 79.70°E | 4270 | Alpine desert | Sand and gravel |

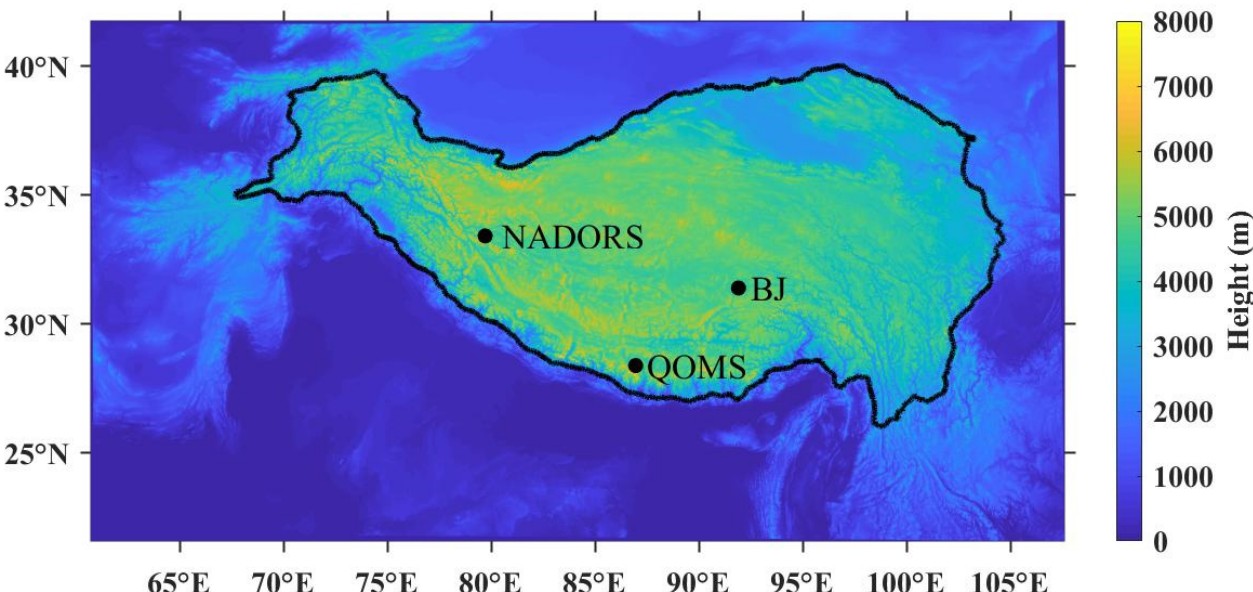

**Figure 1.** Geographical distribution of the observational sites (BJ, QOMS, and NADORS) over the TP (the colored shading denotes topography above sea level).

### 2.2. Observations

The soil data used in this study, measured at the period of 2014–2016, are shared by Ma et al. [52] at the Science Data Bank (https://doi.org/10.11922/sciencedb.00103 (accessed on 26 August 2021), Ma et al. [52]), and the National Tibetan Plateau Data Center (https://doi.org/10.11888/Meteoro.tpdc.270910 (accessed on 26 August 2021)). The data

were hourly values. The information of soil sensors used at each site is presented in Table 2. Note that at 2014 BJ the soil temperature and moisture were measured at the depths of 0.04 m, 0.10 m and 0.20 m, and 0.04 m and 0.20 m, respectively. Both the soil temperature and moisture were measured at the depths of 0.05 m, 0.10 m, and 0.20 m at 2015–2016 BJ.

**Table 2.** The sensors and observation information at each site.

| Site | Variables | Senser Models | Manufactures | Depths | Accuracy | Units |
|------|-----------|---------------|--------------|--------|----------|-------|
| BJ | Tsoil | TS-301/TR-219L | Okazak/ Tri-Tronics | 0.04/0.10/0.20 m; 0.05/0.10/0.20 m | Unknown | °C |
| | θ | CS616-L | Campbell | 0.04/0.20 m; 0.05/0.10/0.20 m | $\pm2.5\%$ θ | $m^3\ m^{-3}$ |
| QOMS | Tsoil | Model 107 | Campbell | 0.0/0.10/0.20 m | $\leq\pm0.01$ °C | °C |
| | θ | CS616 | Campbell | 0.0/0.10/0.20 m | $\pm2.5\%$ θ | $m^3\ m^{-3}$ |
| NADORS | Tsoil | CSI 109 | Campbell | 0.0/0.20 m | $\leq\pm0.03$ °C | °C |
| | θ | CS616 | Campbell | 0.0/0.20 m | $\pm2.5\%$ θ | $m^3\ m^{-3}$ |

The NDVI data were obtained from MYD13C2-v006, provided on a monthly basis, with a spatial resolution of $0.05° \times 0.05°$ (https://giovanni.gsfc.nasa.gov/giovanni/, accessed on 15 April 2022).

### 2.3. The Method Used to Determine Soil Apparent Thermal Diffusivity

Expanding the heat conduction equation presented by Van Wijk and De Vries [30], Gao et al. [23] presented the conduction and convection heat transfer equation with an assumption that the soil apparent thermal diffusivity was vertically homogenous, as follows:

$$\frac{\partial T}{\partial t} = k\frac{\partial^2 T}{\partial z^2} + W\frac{\partial T}{\partial z} \tag{1}$$

where $T$ (°C) is soil temperature, $t$ (s) is the time, and $z$ (m) is the vertical coordinate positive downward; $k$ ($m^2\ s^{-1}$) is soil apparent thermal diffusivity, $W$ ($m\ s^{-1}$) is the apparent convection parameter. With a boundary condition described with the sine function of soil temperature, Gao et al. [23] obtained an analytical solution of this heat transfer equation, and derived the equations of k, as follows:

$$k = -\frac{(z_1 - z_2)^2 \omega \ln(A_1/A_2)}{(\Phi_1 - \Phi_2)\left[(\Phi_1 - \Phi_2)^2 + \ln^2(A_1/A_2)\right]} \tag{2}$$

where $z_1$ (m) and $z_2$ (m) are the measurement depths of soil temperature; $A_1$ (°C) and $A_2$ (°C) are soil temperature amplitude at the depths of $z_1$ and $z_2$, respectively, $\Phi_1$ (rad) and $\Phi_2$ (rad) are soil temperature phase at the depths of $z_1$ and $z_2$, respectively; $\omega$ ($=2\pi/P = 7.292 \times 10^{-5}\ rad^{-1}$) is the angular velocity of the Earth's rotation; and P ($=24 \times 3600$ s) is the harmonic period of the soil temperature.

### 2.4. Data Processing

Equation (2) is the conduction–convection method for determining k, which is the same equation derived by McCallum et al. [53] and Luce et al. [54] for saturated soil. In order to determine k at various timescales, we first derived hourly A and $\Phi$ of daily soil temperature at two depths using the DHR (See Appendix A) from the Captain toolbox [50,51], and then put them into Equation (2) to obtain hourly k. The daily and monthly values of k were obtained by averaging the hourly values.

To ensure the quality of k, the first 3 days of soil k data from the beginning and end of the data collection periods were discarded due to the edge effects of digital filtering with DHR [55,56]. Gordon et al. [57] suggested that data from any time series that have

amplitudes below the sensor resolution should be treated with suspicion. To minimize the amount of suspicion, we deleted the data when the soil temperature amplitude at 0.20 m depth was below the values of 0.5 °C, 0.1 °C and 0.1 °C at BJ, QOMS and NADORS, respectively. After data deletions, approximately 95–100%, 96–100%, and 78–82% of the original data remained for analyses at BJ, QOMS and NADORS, respectively.

The soil moisture measured at the 0.10 m depth was used to represent the soil water status in the 0.0 m to 0.20 m layer. The θ measured at the 0.04 m depth was used as the soil water condition in the interest layer for 2014 BJ. For NADORS, the θ at the 0.10 m depth was calculated by the arithmetic mean of θ measured at the surface and the 0.20 m depth. Note that here θ indicates liquid water content in the soil, and the ice content is not measured. Ice content is qualitatively discussed based on soil temperature and initial soil liquid water content as described in the Section 4.2.

## 3. Results

### 3.1. The Variations of Soil Moisture

Figure 2 shows the variation of soil moisture (θ) over 2014–2016 at the BJ, QOMS, and NADORS. At BJ, θ varied greatly for most months except autumn and winter (October–December) 2015 and winter and spring (January–May) 2016. During non-winter periods, θ varied with rainfall; while during other periods, θ fluctuations were mainly attributed to soil thawing and freezing processes with temperature (e.g., spring (March–April), 2014; autumn (November), 2016). Soil temperatures fluctuated around the freezing temperature (Figure S1), resulting in a water phase transition between liquid and ice. The minimum θ values were similar for 2014–2016, around 0.05 m$^3$ m$^{-3}$ at BJ, while the maximum θ values differed for the three years and occurred in summer, with about 0.33 m$^3$ m$^{-3}$ in July–August 2014, 0.28 m$^3$ m$^{-3}$ in August 2015, and 0.25 m$^3$ m$^{-3}$ in July 2016, respectively.

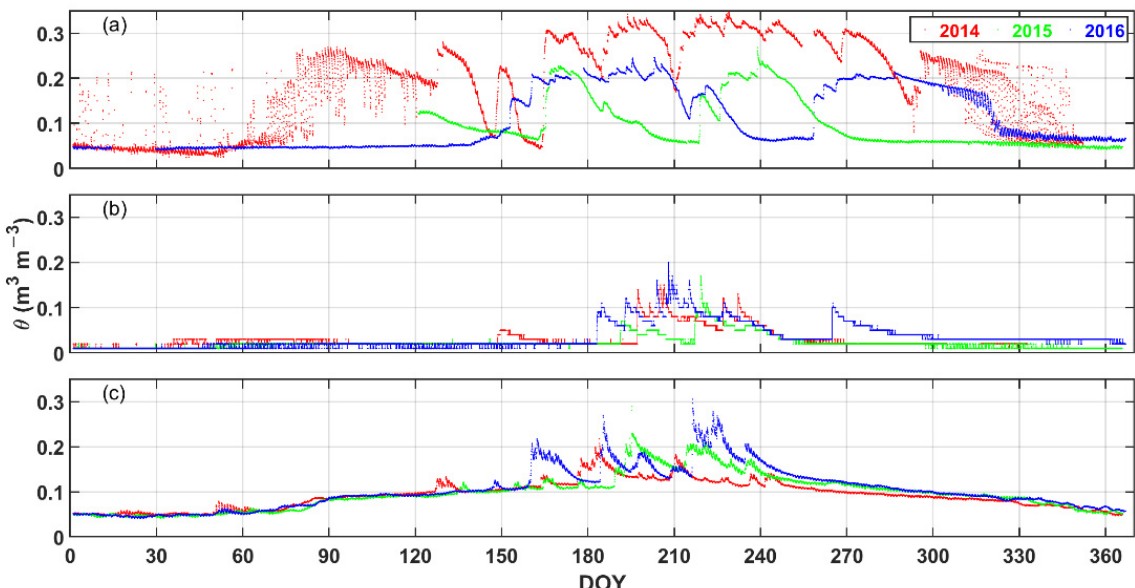

**Figure 2.** The variations of hourly soil moisture (θ, m$^3$ m$^{-3}$) at the depth of 0.10 m at (**a**) BJ, (**b**) QOMS, and (**c**) NADORS for 2014–2016. Note that θ was measured at 0.04 m depth at BJ 2014, θ at NADORS were the arithmetic mean of soil moisture measured at the depths of 0.00 m and 0.20 m, and θ were directly measured at the 0.10 m depth for other cases.

Compared to BJ, the soil at QOMS and NADORS had distinct wet and dry cycles. The θ varied greatly in the summer and remained relatively constant during other periods. Besides, no large fluctuations in the θ were measured in winter at the two sites. The reason was that the θ decreased to a low value before winter, therefore, no large fluctuations in the

θ occurred as soil temperature varied around the freezing temperature. The θ ranged from $0.01$ m$^3$ m$^{-3}$ to $0.20$ m$^3$ m$^{-3}$ at QOMS, and $0.05$ m$^3$ m$^{-3}$ to $0.30$ m$^3$ m$^{-3}$ at NADORS.

Overall, the soil at QOMS was driest, followed by NADORS and BJ. Increases in the θ were sharp after rain, and decreases in the θ were relatively slow as soil water evaporated.

### 3.2. The Variations of Soil Apparent Thermal Diffusivity

The hourly soil temperature amplitude (A) and phase (Φ) at two depths were extracted from soil temperature time series using DHR. For simplicity, only the 2014 results at the three sites are presented as illustrated in Figure 3, and the other results are shown in Figures S2–S4. The data were removed as A at 0.20 m depth was less than the corresponding threshold value at each site.

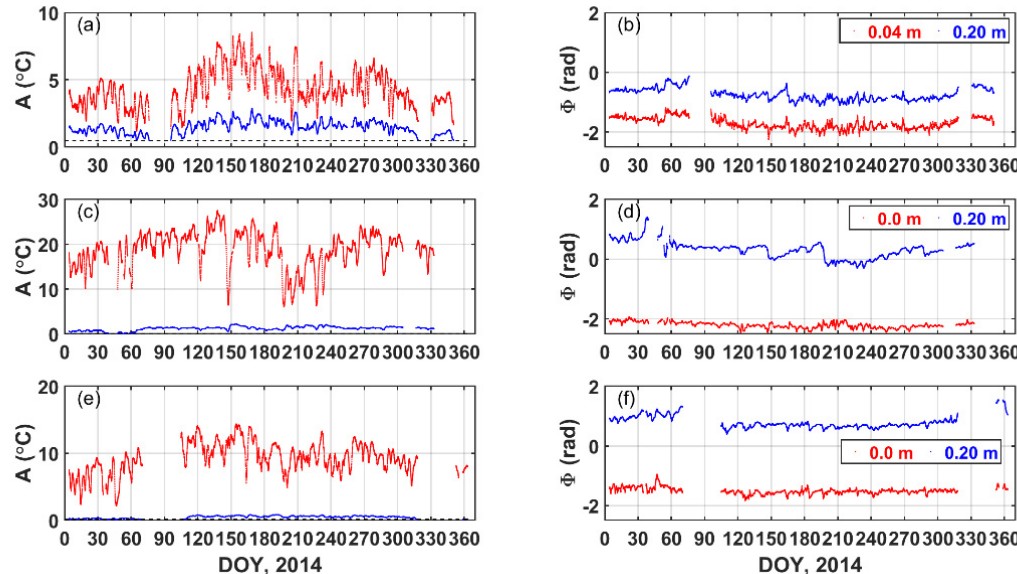

**Figure 3.** The variations of hourly soil temperature amplitude (A, °C) and phase (Φ, rad) at two depths of 0.04 m (or 0.0 m) and 0.2 m at (**a**,**b**) BJ, (**c**,**d**) QOMS, and (**e**,**f**) NADORS for 2014.

It is obvious that A varied with time, and its fluctuation at the shallow depth was much larger than that at the deeper depth (Figure 3a,c,e). As soil is heated by solar radiation at daytime and soil temperature generally decreases exponentially with depth [21], as does A. Compared to other periods, A was relatively small in winter and wet periods. Compared with A, the fluctuation of Φ was relatively small, and the deeper the depth, the larger the Φ. Since soil temperature phase shifts vary linearly with depth [21].

Among the sites, the differences in A and Φ at the two depths were larger at QOMS and NADORS than those at BJ. The soils at QOMS and NADORS were dry (corresponding to relatively small k, see the following results). Thus, soil temperature changes were transmitted slowly through the soil, resulting in large differences in A and Φ between the two depths at the two sites. Tong et al. [58] derived the relationship between conduction–convection k and $\ln(A_1/A_2)$ and $(\Phi_2 - \Phi_1)$ by taking partial derivatives, finding that when $\ln(A_1/A_2)$ is constant, k increases as $(\Phi_2 - \Phi_1)$ decreases; when $(\Phi_2 - \Phi_1)$ is constant, k increases (decreases) with increasing $\ln(A_1/A_2)$ when $(\Phi_2 - \Phi_1) > \ln(A_1/A_2)$ $[(\Phi_2 - \Phi_1) < \ln(A_1/A_2)]$.

After obtaining hourly A and Φ values for soil temperature at the first depth and the 0.20 m depth, hourly k was determined with Equation (2). The daily k was also obtained by averaging the hourly values over a day. The variations of the hourly and daily k for 2014–2016 at the three sites are shown in Figure 4.

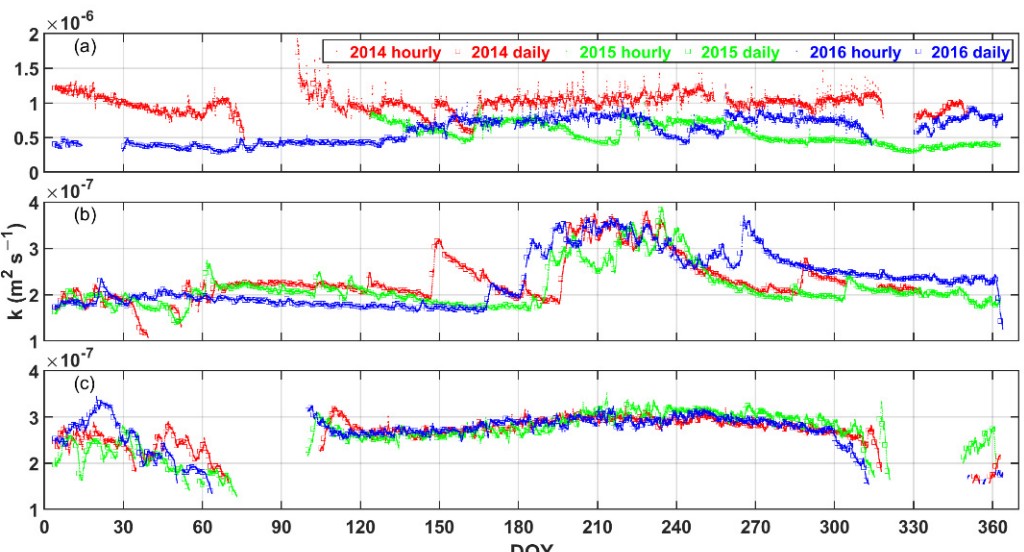

**Figure 4.** The variations of hourly (marked with dot) and daily (marked with square) soil apparent thermal diffusivity (k, $m^2$ $s^{-1}$) of the 0.0 m to 0.20 m layer at (**a**) BJ, (**b**) QOMS, and (**c**) NADORS for 2014–2016.

Generally, k varied with time to varying degrees at different sites and years. At BJ and QOMS, k had an obvious seasonal variation (Figure 4a,b), and it roughly varied with θ during the wetting periods (compare Figure 2a,b to Figure 4a,b).

At BJ, k values in 2014 were larger than those in 2015 and 2016 except for some time in June. The 2014 k varied around $1.0 \times 10^{-6}$ $m^2$ $s^{-1}$, and the minimum and maximum values were about $0.5 \times 10^{-6}$ $m^2$ $s^{-1}$ and $1.9 \times 10^{-6}$ $m^2$ $s^{-1}$, respectively. For 2015–2016, the k ranged from $0.3 \times 10^{-6}$ $m^2$ $s^{-1}$ to $0.9 \times 10^{-6}$ $m^2$ $s^{-1}$. For QOMS, k fluctuated between $1.0 \times 10^{-7}$ $m^2$ $s^{-1}$ to $4.0 \times 10^{-7}$ $m^2$ $s^{-1}$ over 2014–2016, and except for summer time, k varied slightly and was relatively small at most time. The k varied almost exclusively with θ, namely, large k corresponded to large θ (compare Figure 2b to Figure 4b).

Compared to BJ and QOMS, the differences in k at NADORS among the three years were relatively small, and k varied slightly during the non-winter periods, although the θ obviously varied. From January to mid-March, k tended to decrease from $3.3 \times 10^{-7}$ $m^2$ $s^{-1}$ to $1.2 \times 10^{-7}$ $m^2$ $s^{-1}$; while it was relatively stable in spring–mid autumn (May-October), ranging from $2.5 \times 10^{-7}$ $m^2$ $s^{-1}$ to $3.2 \times 10^{-7}$ $m^2$ $s^{-1}$.

Figure 4 shows that k is not always constant throughout a day, and it can change drastically when the soil is wetted (e.g., DOY 218 in 2015 at BJ; DOY 150 in 2014 at QOMS).

The monthly variations of k and θ for 2014–2016 at the three sites are further examined in Figure 5. The values of the monthly k (mean ± one standard deviation) are listed in Table 3.

At BJ, the monthly median k fluctuated around $1.0 \times 10^{-6}$ $m^2$ $s^{-1}$ in 2014. While the 2015 monthly k was around $7.0 \times 10^{-7}$ $m^2$ $s^{-1}$ in May–September and decreased to about $4.0 \times 10^{-7}$ $m^2$ $s^{-1}$ in October–December, the monthly k was around $4.0 \times 10^{-7}$ $m^2$ $s^{-1}$ in February–May and increased to $7.5 \times 10^{-7}$ $m^2$ $s^{-1}$ after May in 2016. For QOMS, the monthly k peaked in August 2014-2015, with a median value of $3.2 \times 10^{-7}$ $m^2$ $s^{-1}$, and varied around $2.0 \times 10^{-7}$ $m^2$ $s^{-1}$ for most other months. In July and August 2016, the monthly k value was the largest, at $3.2 \times 10^{-7}$ $m^2$ $s^{-1}$, after which it decreased almost linearly until December ($2.3 \times 10^{-7}$ $m^2$ $s^{-1}$). The monthly k was relatively stable at around $1.8 \times 10^{-7}$ $m^2$ $s^{-1}$ before July.

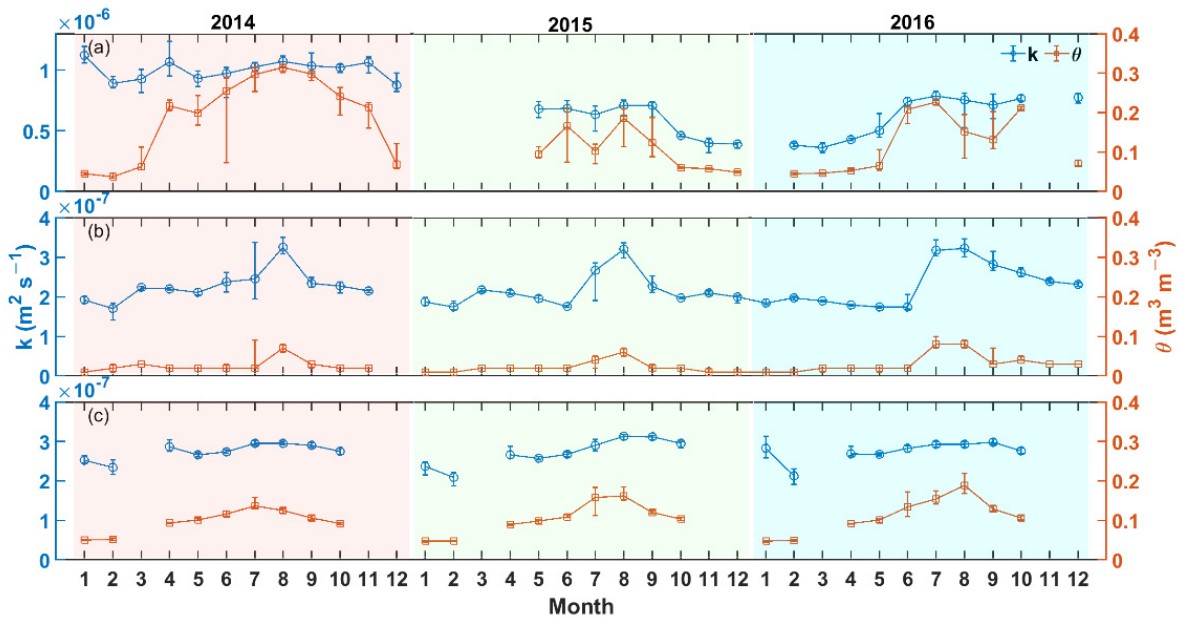

**Figure 5.** The monthly variations (25th, median, and 75th percentiles) of soil apparent thermal diffusivity (k, m² s⁻¹) and moisture (θ, m³ m⁻³) at (**a**) BJ, (**b**) QOMS, and (**c**) NADORS for 2014–2016. Data are deleted when there are less than 15 days available in a month.

**Table 3.** The monthly (mean ± one standard deviation) $k \times 10^7$ (m² s⁻¹) at each site.

| Month | BJ | | | QOMS | | | NADORS | | |
|---|---|---|---|---|---|---|---|---|---|
| | 2014 | 2015 | 2016 | 2014 | 2015 | 2016 | 2014 | 2015 | 2016 |
| 1 | 11.2 ± 0.8 | - [1] | - | 1.9 ± 0.1 | 1.9 ± 0.1 | 1.9 ± 0.1 | 2.6 ± 0.1 | 2.3 ± 0.2 | 2.9 ± 0.3 |
| 2 | 9.0 ± 0.6 | - | 3.8 ± 0.3 | 1.7 ± 0.3 | 1.8 ± 0.2 | 1.9 ± 0.1 | 2.4 ± 0.2 | 2.1 ± 0.3 | 2.2 ± 0.3 |
| 3 | 9.0 ± 1.4 | - | 3.7 ± 0.5 | 2.2 ± 0.1 | 2.2 ± 0.1 | 1.9 ± 0.0 | - | - | - |
| 4 | 11.2 ± 2.3 | - | 4.3 ± 0.2 | 2.2 ± 0.0 | 2.1 ± 0.1 | 1.8 ± 0.0 | 2.9 ± 0.2 | 2.7 ± 0.2 | 2.8 ± 0.2 |
| 5 | 9.1 ± 1.1 | 6.9 ± 0.8 | 5.2 ± 1.0 | 2.2 ± 0.3 | 2.0 ± 0.0 | 1.7 ± 0.0 | 2.7 ± 0.1 | 2.6 ± 0.1 | 2.7 ± 0.1 |
| 6 | 9.3 ± 1.6 | 6.5 ± 1.2 | 7.4 ± 0.7 | 2.4 ± 0.3 | 1.8 ± 0.0 | 1.9 ± 0.0 | 2.8 ± 0.1 | 2.7 ± 0.1 | 2.8 ± 0.1 |
| 7 | 10.3 ± 0.9 | 6.1 ± 1.2 | 7.9 ± 0.6 | 2.7 ± 0.7 | 2.5 ± 0.5 | 3.2 ± 0.3 | 3.0 ± 0.1 | 2.9 ± 0.2 | 2.9 ± 0.1 |
| 8 | 10.8 ± 0.9 | 6.7 ± 1.3 | 6.9 ± 1.5 | 3.3 ± 0.2 | 3.2 ± 0.3 | 3.2 ± 0.3 | 3.0 ± 0.1 | 3.1 ± 0.1 | 2.9 ± 0.1 |
| 9 | 10.6 ± 1.0 | 6.7 ± 0.9 | 7.1 ± 1.2 | 2.4 ± 0.2 | 2.3 ± 0.3 | 2.9 ± 0.3 | 2.9 ± 0.1 | 3.1 ± 0.1 | 3.0 ± 0.1 |
| 10 | 10.1 ± 0.7 | 4.6 ± 0.2 | 7.7 ± 0.5 | 2.4 ± 0.2 | 2.0 ± 0.0 | 2.6 ± 0.1 | 2.8 ± 0.1 | 2.9 ± 0.1 | 2.7 ± 0.2 |
| 11 | 10.3 ± 1.2 | 3.9 ± 0.6 | - | 2.2 ± 0.0 | 2.1 ± 0.0 | 2.4 ± 0.0 | - | - | - |
| 12 | 8.8 ± 0.8 | 3.8 ± 0.2 | 7.6 ± 0.8 | - | 1.9 ± 0.1 | 2.3 ± 0.0 | - | - | - |

[1] indicates there is no data.

Differing from BJ and QOMS, the monthly k at NADORS varied little, about $3.0 \times 10^{-7}$ m² s⁻¹ for most of the non-winter period (April-October), although θ did vary during this period. It decreased from January to February over the 3-year period, ranging from $3.0 \times 10^{-7}$ m² s⁻¹ to $1.0 \times 10^{-7}$ m² s⁻¹.

Overall, the variation trends of monthly k were roughly similar to those of θ, except for BJ 2014 and NADORS 2016. The monthly k at BJ ranged from $0.4(\pm 0.0) \times 10^{-6}$ m² s⁻¹ to $1.1(\pm 0.2) \times 10^{-6}$ m² s⁻¹, from $1.7(\pm 0.0) \times 10^{-7}$ m² s⁻¹ to $3.3(\pm 0.2) \times 10^{-7}$ m² s⁻¹ at QOMS, and from $2.1(\pm 0.3) \times 10^{-7}$ m² s⁻¹ to $3.1(\pm 0.1) \times 10^{-7}$ m² s⁻¹ at NADORS (Table 3).

### 3.3. The Relationship between Soil Apparent Thermal Diffusivity and Moisture

Figure 6 shows how k varies with θ on an hourly timescale in 2014–2016 at the sites. Overall, the trends of unfrozen soil k versus θ at the three sites were roughly similar, i.e., k increased rapidly to a maximum value with increasing θ and then tended to be constant or decrease slightly as θ increased further. The values of θ corresponding to peak k values were different, and the θ values at QOMS were less than those at BJ.

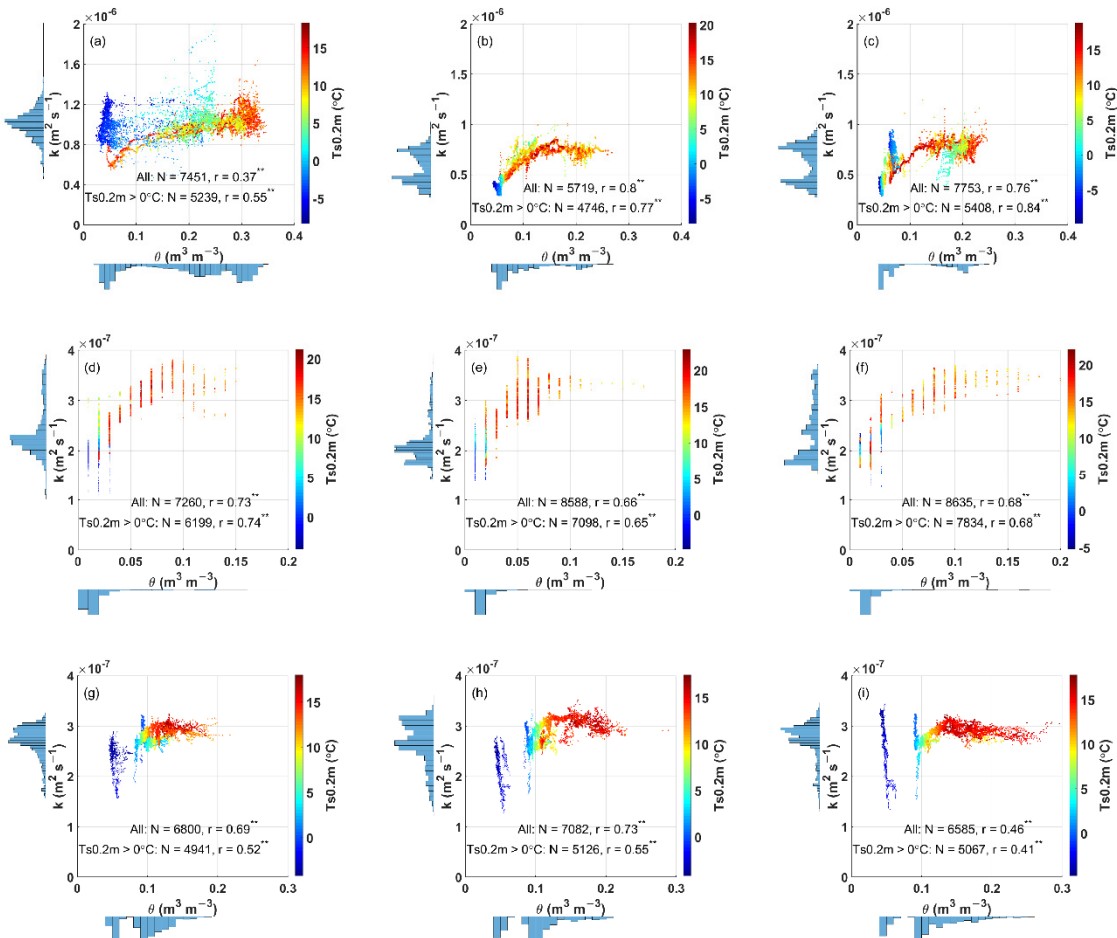

**Figure 6.** The variation of soil apparent thermal diffusivity (k, m$^2$ s$^{-1}$) with soil moisture (θ, m$^3$ m$^{-3}$) at 0.10 m depth on an hourly timescale for 2014 (in the 1st column), 2015 (in the 2nd column) and 2016 (in the 3rd column) at (**a–c**) BJ, (**d–f**) QOMS, and (**g–i**) NADORS, respectively. The color bar indicates the value of soil temperature at the depth of 0.20 m (Ts0.2m), and the larger the temperature, the redder the color. The marker size suggests the amplitude of Ts0.2m (A0.2m), and the larger the value of A0.2m, the larger the marker. The correlation coefficients (r) between k and θ are provided for all of the data and for the data when Ts0.2m > 0 °C, respectively. "**" indicates *p*-value < 0.01. The probability distributions of k and θ are shown on the y-axis and x-axis sides, respectively.

Interestingly, the relationship between k and θ did not appear to be significant when Ts0.2m < 0 °C and θ < 0.1 m$^3$ m$^{-3}$, i.e., k fluctuated greatly within a narrow range of θ (e.g., see the blue points in Figure 6a,c,g–i). The correlation coefficients (r) between k and θ for all of the data ranged from 0.66 to 0.80 except for BJ 2014 (r = 0.37) and NADORS 2016 (r = 0.46). Under the condition of Ts0.2m > 0 °C, the r coefficients changed to different degrees at each site and year, and the changes depended on the location of the data of Ts0.2m ≤ 0 °C on the curve. Overall, without including data for Ts0.2m ≤ 0 °C, the r coefficients decreased at NADORS, increased at BJ, and changed slightly at QOMS, respectively.

Note that the probability distributions of k were different, especially at different sites. The same is true for θ. Compared to k, the probability distribution of θ for a given site was more consistent during 2014–2016. There was an obvious gap between 0.05 m$^3$ m$^{-3}$ to 0.10 m$^3$ m$^{-3}$ at NADORS (Figure 6g–i), since the data in the θ range were deleted according to the standard of A0.2m < 0.1 °C, as mentioned in Section 2.4.

The relationship between k and θ on a daily timescale (Figure S5) was similar to that on an hourly timescale. Few studies have investigated the relationship between k and θ for soil below and above the freezing temperature simultaneously.

We furthermore investigated the relationship between median k and θ on a monthly timescale, as shown in Figure 7. Similar to the hourly results, the monthly k tended to increase with θ when θ was relatively small, reached a peak value and then became relatively stable as θ increased further.

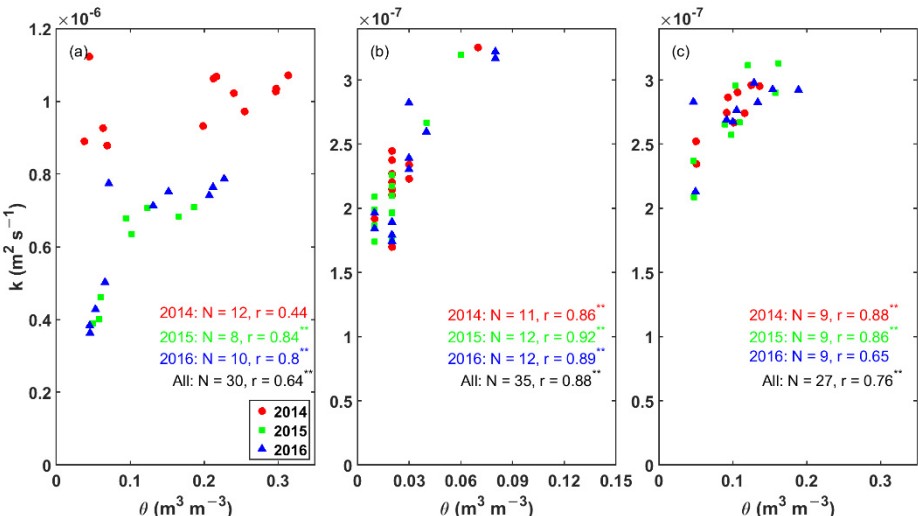

**Figure 7.** Same as Figure 6, but for the relationship between median k and θ on a monthly timescale. The number of months per year (N) and corresponding correlation coefficients (r) are given. "**" indicates *p*-value < 0.01, and there is no label after r if r is not significant (*p*-value > 0.05). The data for a month are deleted when the number of days is less than 15.

Overall, monthly k was significantly correlated to θ regardless of site and year, except for 2014 BJ and 2016 NADORS. The significant (*p*-value < 0.01) r ranged from 0.80 (2016 BJ) to 0.92 (2015 QOMS) on a monthly timescale, and from 0.64 (NADORS) to 0.88 (QOMS) on an annual timescale. The r coefficients of k vs. θ on a monthly timescale were larger than those on an hourly timescale. This could be explained because the effect of frozen soil on the relationship between k and θ was greatly reduced on a monthly timescale.

In order to indirectly investigate the effect of soil moisture on k from the vegetation aspect, the relationship between monthly k and NDVI is also examined as shown in Figure 8. The ranges of NDVI were different at the three sites, and the maximum NDVI at BJ (0.51) was approximately 2.5 times that of QOMS (0.22) and NADORS (0.18).

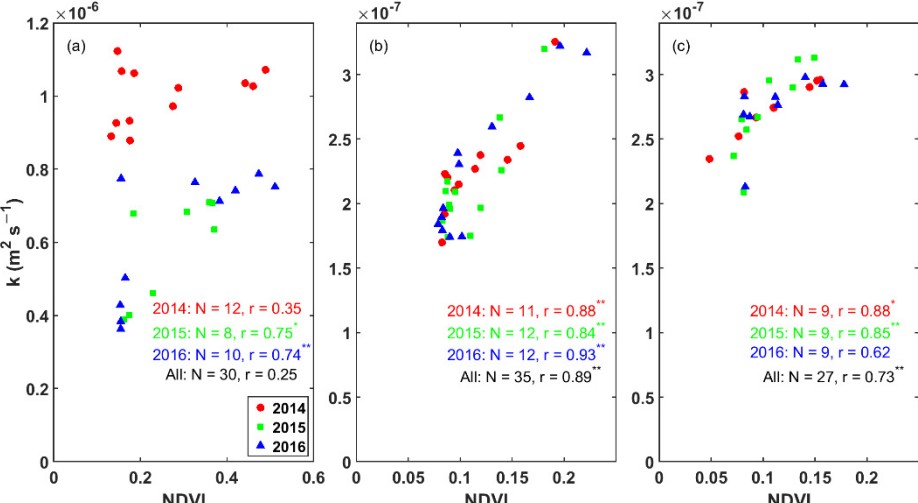

**Figure 8.** Same as Figure 7, but for the relationship between k and NDVI. "*" indicates *p*-value < 0.05, "**" indicates *p*-value < 0.01, and there is no label after r if r is not significant (*p*-value > 0.05).

Interestingly, the monthly k had a similar correlation with NDVI as it did with θ. At QOMS and NADORS, the r coefficients of k vs. NDVI were close to those of k vs. θ (Δr < 0.08), while the r coefficients of k vs. NDVI were smaller than those of k vs. θ at BJ.

## 4. Discussion

### 4.1. The Variations of Soil Apparent Thermal Diffusivity

With k methods based on the solution of the soil heat transfer equation, normally only daily or longer timescale k values were obtained in previous studies. In this study, we used a conduction–convection method combined with DHR to obtain hourly k. The daily and monthly k were also provided for 2014–2016 at the three sites (Figures 4 and 5 and Table 3). Figure 4 visually shows that in a day, k is not necessarily constant over a day, and it can change drastically when the soil become wetter suddenly. The knowledge of k with higher temporal resolution may have great implications in improving soil heat and water models. The issue of underestimating soil temperature at night with the conduction-convection method [24] may be largely resolved by using hourly k values instead of daily k values as inputs. Besides, higher temporal resolution k may help to improve the modeling of permafrost distributions [11].

Figure 4 also indicates that k had obvious seasonal variations at BJ and QOMS, while it did not vary greatly during wetting at NADORS. The soil types for both QOMS and NADORS were sandy and gravel, but θ at NADORS during wetting (> 0.10 m$^3$ m$^{-3}$) was greater than that at QOMS (Figure 2b,c). Previous studies suggested that k variations during wetting depended on the magnitude of θ, and k was insensitive to changes in θ when θ reached certain thresholds (e.g., 0.1~0.2 m$^3$ m$^{-3}$ for sand soils, [16]), which could explain why k did not vary much during wetting at NADORS.

The k values at QOMS and NADORS were much less than those at BJ (Figures 4 and 5). In addition to the soil moisture (Figure 2), the soil bulk density at QOMS and NADORS was expected to differ from that at BJ, since their soil types were sand and gravel, whereas BJ's soil type was sandy silt loam. Previous studies indicated that k varied with bulk density as well as soil moisture [17,18,59–61]. Therefore, the combined effects of the soil texture, moisture and bulk density resulted in the relative magnitude of k at the three sites. At BJ, the k values in 2014 were much larger than those at 2015 and 2016 (Figure 6a–c). The main reason may be due to differences in soil bulk density and soil moisture content.

With the conduction–convection method, Gao [26] determined the daily k ranging from 0.1 × 10$^{-6}$ to 2.0 × 10$^{-6}$ m$^2$ s$^{-1}$ at BJ during DOYs 195 to 258, 1998, which was within the range of k at BJ for 2014–2016 in this study. Additionally, with the conduction–convection method, Zhou et al. [28] determined daily k based on soil temperatures measured at 0.8 m and 3.2 m depths in 39 weather sites in the TP during 1980 to 2001. They reported that the magnitude of k in most areas of the TP was 10$^{-7}$ m$^2$ s$^{-1}$, and relatively high k values were obtained at the central and eastern parts of the plateau with an order of magnitude of 10$^{-6}$ m$^2$ s$^{-1}$. Our results that k at BJ are larger than those at QOMS and NADORS are broadly consistent with the findings of Zhou et al. [28]. The soil thermal properties were determined at a cold semi-desert site on the western TP for about two years by Wang et al. [48]. They calculated k as the ratio of $\lambda/C_v$, which were determined by soil temperature and heat fluxes measured at two layers according to the method proposed by Zhang and Huang [49]. Their daily k values ranged from 3.0 × 10$^{-7}$ m$^2$ s$^{-1}$ to 9.0 × 10$^{-7}$ m$^2$ s$^{-1}$, which was larger than our k values at QOMS and NADORS, and less than those at BJ. The differences in k were attributed to the differences in soil texture, moisture, and the method of determining k. Even with a well-calibrated soil heat flux sensor, it is difficult to measure the soil heat flux accurately [62–65], because the soil heat and moisture fluxes are disturbed [66]. Heat flux plates measure only sensible heat as it moves past the plate by means of the temperature gradient, which exists across the plate. Latent heat, which is hidden in the evaporative process, is not detected [67]. Therefore, cautions should be exercised when determining soil thermal properties using soil heat flux plate data.

Using monthly soil temperature data measured in the 0.2 m to 3.2 m soil layer at the northern permafrost regions, Zhu et al. [11] obtained k with a conduction method, ranging from $0.2 \times 10^{-7}$ m$^2$ s$^{-1}$ to $2.0 \times 10^{-6}$ m$^2$ s$^{-1}$. The minimum k values occurred at sites with large soil organic carbon (SOC, 60~70 kg C m$^{-3}$), and they were less than our minimum results. A possible reason is that there is more SOC at the northern permafrost regions than at our sites, and the k of SOC is an order of magnitude smaller than that of typical soil minerals [68].

Few studies provided long-term k using a conduction–convection method on different timescales over the TP. The long-term k values obtained in this study on different timescales can be used as input for land surface models over this region.

### 4.2. The Relationship between Soil Apparent Thermal Diffusivity and Soil Moisture

Normally, k increases rapidly with increasing θ to a maximum and then decreases with further increases in θ. This is explained by the fact that the heat capacity increases linearly with θ, whereas the thermal conductivity experiences its most rapid rise at low θ, leading to the ratio of thermal conductivity to heat capacity to have an internal maximum as a function of θ [29,69]. The increase rate of k with θ differs for different soil textures. A laboratory experiment showed that for sandy soils, k increased rapidly with θ when θ was less than 0.1~0.2 m$^3$ m$^{-3}$, and then it remained stable or slightly decreased; while the variation of k with θ was smaller for silty and clay-textured soils [16].

Figure 6 shows that although the probability distributions of k and θ differed, especially at different sites, the variation trends of unfrozen soil k versus θ were roughly similar, which was consistent with previous studies reported based on laboratory measured data under controlled conditions (e.g., [16,17,29,59]) and in situ data (e.g., [22,25,70]). We should be aware that this trend of k versus θ reported with laboratory measurements generally only applies when the soil temperature is room temperature, since measurements in the laboratory are usually conducted at room temperature. However, if some ice is present in a soil layer, the k versus θ relationship is expected to deviate from this trend, since ice has a higher k value than water (1.1 vs. $0.14 \times 10^{-6}$ m$^2$ s$^{-1}$, [68]). That is why there are some "outliers" in the trend of k versus θ as mentioned above, i.e., k varied greatly within a narrow range of θ, as the blue point when θ < 0.1 m$^3$ m$^{-3}$ shown in Figure 6a,c,g–i.

About 2/3 blue points in Figure 6c (when θ ranged from 0.06 m$^3$ m$^{-3}$ to 0.075 m$^3$ m$^{-3}$) appeared on DOYs 332–364. To elucidate this phenomenon, the variations of soil temperature, θ and k over DOYs 300–365, 2016, at BJ are shown in Figure 9. One can see that on DOYs 332–364 soil liquid water content, θ in the 0.05–0.20 m layer showed a downward trend, and as the soil temperature dropped to the freezing point, more and more ice was expected to form. At the same time, θ at the depth of 0.10 m varied slightly, while k tended to increase greatly. Therefore, one can see that k had a wide range from $0.6 \times 10^{-6}$ m$^2$ s$^{-1}$ to $0.95 \times 10^{-6}$ m$^2$ s$^{-1}$ with a narrow range of θ in Figure 6c. Note that almost no water loss is assumed since DOY 300 here, as soil water evaporation is generally small when soil temperature is below freezing. Therefore, the decrease in liquid water content in the soil layer is expected to be a result of increasing ice content.

For Figure 6i, the "outliers" (when θ < 0.054 m$^3$ m$^{-3}$ and k ranged from $1.4 \times 10^{-7}$ m$^2$ s$^{-1}$ to $3.4 \times 10^{-7}$ m$^2$ s$^{-1}$) appeared on DOYs 1–63, 2016 at NADORS. The variations of soil temperature, θ and k over DOYs 1–70, 2016, at NADORS are shown in Figure 10. In contrast to Figure 9, the soil temperature tended to increase from DOYs 1 to 63, resulting in an increasing amount of time that soil temperature was above freezing (Figure 10a). We expect that ice content in the 0–0.20 m soil layer tended to decrease over time, although θ appeared to vary slightly. Figure 10 also shows that as the ice content decreased with increasing soil temperature, k tended to decrease, eventually varying from approximately $1.4 \times 10^{-7}$ m$^2$ s$^{-1}$ to $3.4 \times 10^{-7}$ m$^2$ s$^{-1}$. Therefore, one can see that k varied greatly with a narrow range of θ in Figure 6i. Figures 9 and 10 explains why large k changes occurred over a small range of θ in Figure 6a,c,g–i.

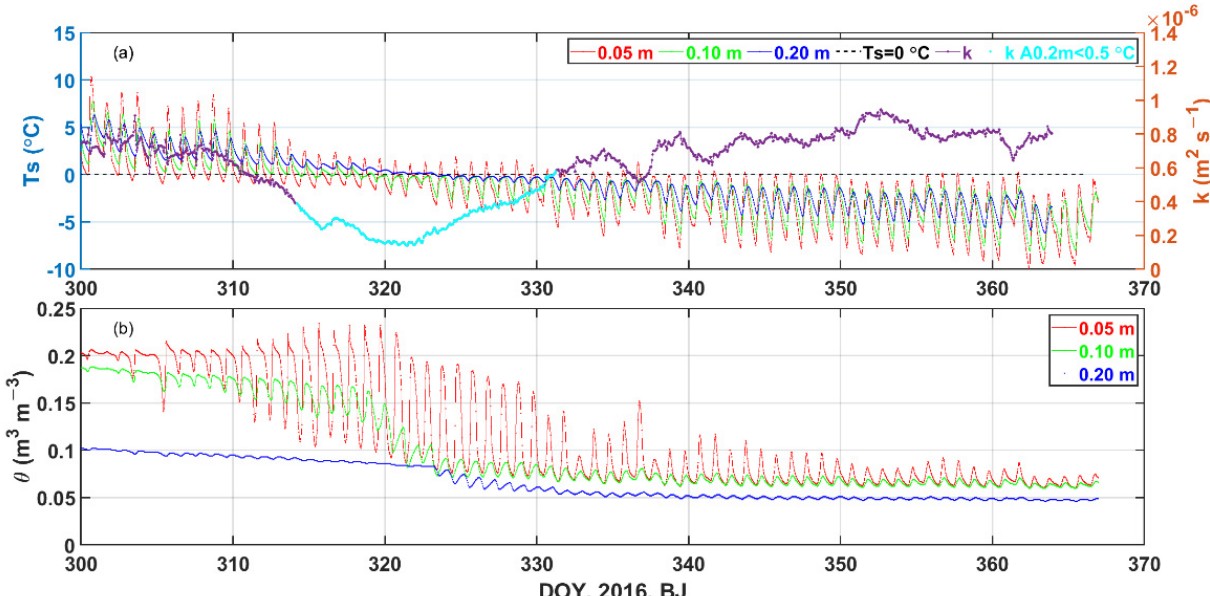

**Figure 9.** The (**a**) soil temperature (Ts, °C) at the depths of 0.05 m, 0.10 m and 0.20 m, and soil apparent thermal diffusivity (k, m² s⁻¹), and (**b**) soil moisture ($\theta$, m³ m⁻³) at the depths of 0.05 m, 0.10 m and 0.20 m over DOYs 300–365, 2016, at BJ. The labels of k are marked in cyan when the soil amplitude at the 0.2 m depth is less than 0.5 °C.

Compared to BJ and NADORS, there were no apparent "outliers" at QOMS (Figure 6d–f). The soil at QOMS was driest among the three sites, and $\theta$ decreased to the minimum value (<0.03 m³ m⁻³) in September from the summer (Figure 2b), and small $\theta$ lasted until winter. Some laboratory experiments indicated that the unfrozen water content of freezing soil was largely controlled by the initial volumetric water content [71], which may prove why there was almost no ice after September at QOMS. Ochsner and Baker [20] presented some in situ measurements of soil thermal properties across a full range of temperatures encountered in freezing and thawing soil, and the measurement and model both showed that for temperatures between −5 °C and 0 °C, soil thermal properties were strongly temperature dependent. They explained that temperature dependence was primarily the result of latent heat transfer processes when water underwent a phase change. Therefore, although soil temperature changed from positive to negative at QOMS in autumn, the "frozen" soil had a lesser effect on k since little ice was produced. Thus, few "outliers" existed during the cold months at QOMS (Figure 6d–f).

To indirectly investigate the effect of soil moisture on k from the vegetation aspect, the relationship between monthly k and NDVI was also examined (Figure 8), which is similar to the relationship between k to $\theta$ (Figure 7). The reason may be that NDVI is closely related to $\theta$ at these sites. The r coefficients between NDVI and $\theta$ ranged from 0.73 to 0.92 on a monthly timescale, and from 0.69 to 0.82 on an annual timescale (Table 4). Compared to the other two sites, more vegetation was present in BJ, so the NDVI of BJ represented more vegetation information than bare soil. Therefore, the correlation between k and NDVI was weaker than that between k and $\theta$ at BJ, while closer to that between k and $\theta$ at the other two sites, as shown in Figures 7 and 8.

With in situ soil temperature, we determined k by the conduction–convection method for 2014–2016 at three TP sites and examined the relationship between k and $\theta$. Our results indicated that k had a clear relationship with $\theta$ for unfrozen soil, but the relationship changed when the soil temperature was less than 0 °C and the initial $\theta$ was not too small. These findings broaden our understanding of the relationship between in situ k and $\theta$.

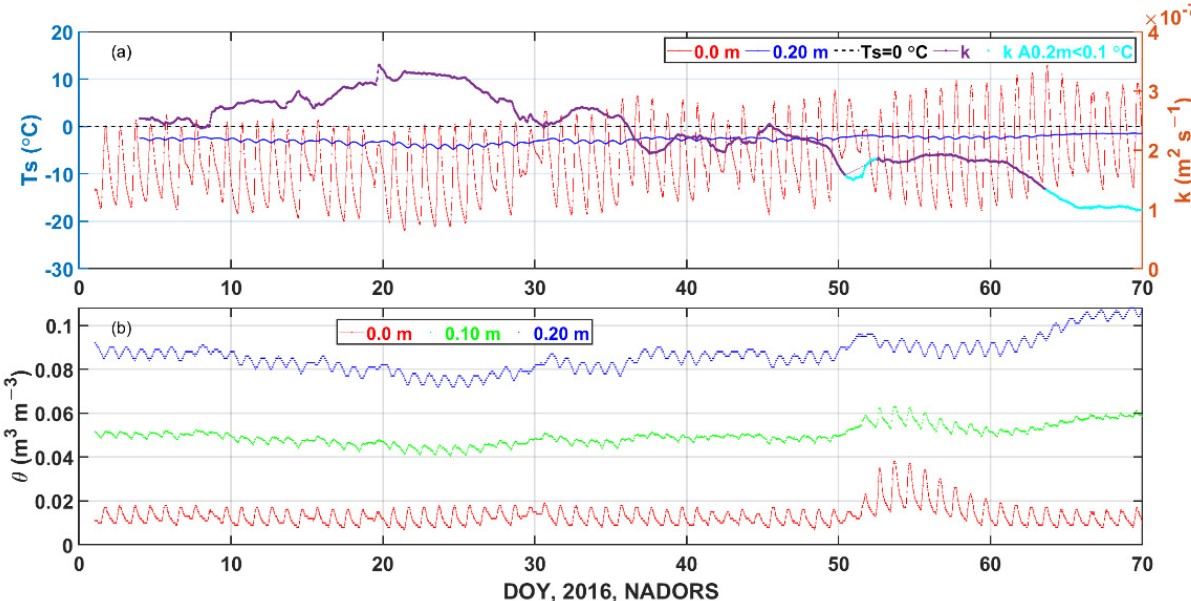

**Figure 10.** The (**a**) soil temperature (Ts, °C) at the depths of 0.0 m and 0.20 m, and soil apparent thermal diffusivity (k, m² s⁻¹), and (**b**) soil moisture (θ, m³ m⁻³) at the depths of 0.0 m, 0.10 m and 0.20 m over DOYs 0–70, 2016, at NADORS. The labels of k are marked in cyan when the soil amplitude at the 0.2 m depth is less than 0.1 °C.

**Table 4.** Correlation coefficient (r) of the monthly soil moisture and NDVI for 2014-2016 at the three sites.

| Site | 2014 | 2015 | 2016 | All |
|---|---|---|---|---|
| BJ | 0.82 ** [1] | 0.75 * [2] | 0.86 ** | 0.69 ** |
| QOMS | 0.73 * | 0.87 ** | 0.92 ** | 0.86 ** |
| NADORS | 0.87 ** | 0.88 ** | 0.91 ** | 0.86 ** |

[1] ** *p*-value < 0.01, [2] * *p*-value < 0.05.

### 4.3. Limitations

In this study, long-term k was determined for frozen and unfrozen conditions. Due to the uncertainty of the k method, several k values were removed during the transition periods with soil thawing/freezing when soil temperature variations were low, as mentioned in Section 2.4. Another method or sensor is needed to determine k during soil thawing/freezing periods.

Our results indicated a potential effect of ice content on the relationship between k and θ, while the amount of ice content was inferred by combining the variations of soil temperature and soil moisture, rather than direct measurements. The ice content measured in situ is vital to quantify the relationship between k and soil moisture (including liquid and ice phases). The thermo-TDR sensor is a candidate for in situ measurement of both liquid and ice contents, since its performance was satisfactory in laboratory experiments reported in previous studies (e.g., [72–74]). In addition, thermo-TDR was also used in the field to measure soil thermal properties during thawing and freezing (e.g., [20]). By using actively heated fiber Bragg grating (AH-FBG) sensors, Wu et al. [75] measured the ice content of frozen soil in laboratory. The AH-FBG sensor integrates the functions of active heating and temperature measurement, which can accurately detect the thermal response of frozen soil [75]. We recommend using soil thermocouples and thermo-TDR sensors or only AH-FBG sensors for soil temperature, water content (liquid and ice phase) and thermal property measurements over multiple thawing and freezing cycles to more deeply explore the time variations of k and its relationship with water content.

In this study, we did not examine the effects of other soil factors (e.g., soil texture, SOC) on k due to a lack of data. Zhu et al. [11] suggested that SOC is the dominate factor (among soil texture, bulk density, moisture, and SOC) controlling the variability of diffusivity at 200 sites in high latitude regions, and k is a strong predictor for simulated permafrost extent. In future investigations, additional soil factors should be included in the study of long-term variations of soil thermal properties.

## 5. Conclusions

Based on in situ soil temperature data measured at three TP sites (BJ, QOMS, and NADORS), we determined the hourly, daily, and monthly soil apparent thermal diffusivity values of the 0.0 m to 0.20 m layer for 2014–2016 by using a conduction–convection method combined with DHR. The hourly, daily, and monthly k values of the 0.0 m to 0.20 m layer were obtained. The hourly and daily k values ranged from $0.3 \times 10^{-6}$ m$^2$ s$^{-1}$ to $1.9 \times 10^{-6}$ m$^2$ s$^{-1}$ at BJ, and from $1.0 \times 10^{-7}$ m$^2$ s$^{-1}$ to $4.0 \times 10^{-7}$ m$^2$ s$^{-1}$ at QOMS and NADORS. The monthly k ranged from $0.4(\pm 0.0) \times 10^{-6}$ m$^2$ s$^{-1}$ to $1.1(\pm 0.2) \times 10^{-6}$ m$^2$ s$^{-1}$ at BJ, from $1.7(\pm 0.0) \times 10^{-7}$ m$^2$ s$^{-1}$ to $3.3(\pm 0.2) \times 10^{-7}$ m$^2$ s$^{-1}$ at QOMS, and from $2.1(\pm 0.3) \times 10^{-7}$ m$^2$ s$^{-1}$ to $3.1(\pm 0.1) \times 10^{-7}$ m$^2$ s$^{-1}$ at NADORS. The results suggested that k was not constant over a day, and k showed seasonal variations. The variations of k with θ appeared to be roughly similar for unfrozen soil at these sites and years, namely, k increased sharply before it reached a peak value as θ increased, and then it tended to be stable or varied slightly with further increases in θ. The correlation coefficients (r) between k and θ ranged from 0.37 to 0.80, and 0.80 to 0.92 on hourly and monthly timescales, respectively. However, the relationship between k and θ changed when soil temperature was below 0 °C. Our results also suggested that the k and NDVI values were significantly related on monthly and annual timescales, with r ranging from 0.73 to 0.93. These results broaden our understanding of the relationship between in situ k and θ. The presented values of k at various timescales can be used as soil parameters when modeling land–atmosphere interactions at these TP regions.

**Supplementary Materials:** The following supporting information can be downloaded at: https://www.mdpi.com/article/10.3390/rs14174238/s1, Figure S1. The variations of soil temperature at (a) BJ, (b) QOMS, (c) NADORS, respectively. Figure S2. The amplitude (A, °C) and phase (Φ, rad) of soil temperature 2015–2016 at BJ. Figure S3. The amplitude (A, °C) and phase (Φ, rad) of soil temperature 2015–2016 at QOMS. Figure S4. The amplitude (A, °C) and phase (Φ, rad) of soil temperature 2015–2016 at NADORS. Figure S5. The variation of soil apparent thermal diffusivity (k, m$^2$ s$^{-1}$) with soil moisture (θ, m$^3$ m$^{-3}$) on a daily timescale in 2014 (in the 1st column), 2015 (in the 2nd column) and 2016 (in the 3rd column) at (a–c) BJ, (d–f) QOMS, and (g–i) NADORS, respectively.

**Author Contributions:** B.T.: formal analysis, writing—original draft, writing—review and editing, funding acquisition. H.X.: formal analysis, writing—review and editing. R.H.: conceptualization, writing—review and editing. L.B.: conceptualization, supervision. J.G.: supervision, funding acquisition, writing—review and editing. All authors have read and agreed to the published version of the manuscript.

**Funding:** This research was funded by the Second Tibetan Plateau Scientific Expedition and Research program (STEP) (2019QZKK0102), the Natural Science Foundation of China (U2142209), and the China Postdoctoral Science Foundation (2021M703558).

**Data Availability Statement:** The data that support the findings of this study are available from the Science Data Bank (https://doi.org/10.11922/sciencedb.00103; Ma et al. 2020 [52]) and additionally at the National Tibetan Plateau Data Center (https://doi.org/10.11888/Meteoro.tpdc.270910).

**Acknowledgments:** We appreciate the access to the NASA datasets. The authors appreciate all of the hard work done by researchers attending the Tibetan Plateau (TP) Experiment.

**Conflicts of Interest:** The authors declare that they have no known competing financial interests or personal relationships that could have appeared to influence the work reported in this paper.

**Appendix A. Determination of Soil Temperature Amplitude and Phase with DHR**

Gordon et al. [57] described how to extract soil amplitude and phase information from a soil temperature time series with the Dynamic Harmonic Regression (DHR). DHR, a simplification of the unobserved component model, has the form as follows:

$$y_t = T_t + C_t + e_t \tag{A1}$$

where $y_t$ is the observed soil temperature time series, $T_t$ is a trend or zero-frequency component, $C_t$ is a cyclical component, and $e_t$ is an irregular, white-noise component [50]. The $C_t$ is modeled as a sum of the fundamental signal and its associated harmonics, as follows:

$$C_t = \sum_{i=1}^{N} [a_{i,t} cos(\omega_i t) + b_{i,t} sin(\omega_i t)] \tag{A2}$$

where $a_{i,t}$ and $b_{i,t}$ are stochastic time-varying parameters and $\omega_i$ (I = 1:N) are the fundamental frequency and its harmonics ($\omega_1$) up to the Nyquist frequency ($\omega_N$). This DHR model is a non-stationary extension of the discrete Fourier transform, where the amplitude (A) and phase ($\Phi$) of the soil temperature for each time series vary with time. Identification of the time-varying parameters is achieved in a stochastic state formulation using two-step Kalman filtering and fixed-interval smoothing [50].

After obtaining the time-varying parameters, the A and $\Phi$ of any harmonic component at discrete time can be calculated by the following equations:

$$A_{i,t} = \sqrt{a_{i,t}^2 + b_{i,t}^2} \tag{A3}$$

$$\Phi_{i,t} = tan^{-1}(a_{i,t}/b_{i,t}) \tag{A4}$$

where $A_{i,t}$ and $\Phi_{i,t}$ are the amplitude and phase for the component with frequency $\omega_i$ at time $t$, respectively.

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
