# Peer review of "Determination of Long-Term Soil Apparent Thermal Diffusivity Using Near-Surface Soil Temperature on the Tibetan Plateau"

_remotesensing, doi:10.3390/rs14174238_

Round 1

Reviewer 1 Report

In this work the authors have tried to Determine the long-term soil apparent thermal diffusivity using near-surface soil temperature on the Tibetan Plateau. The parameter k which represents the soil thermal diffusivity was computed from soil temperature using the conduction-convection method at different time scales. The results from this work have shown the importance of the time scale variation in the computation of the k parameter and for future modelling. However, for the clarity of this work, a number of areas emphasized above require some improvements.

Major Comments 

In this work much emphasis was laid on the soil temperature and K values at freezing and thawing temperature, Particularly at freezing temperature. It is therefore pertinent to check and compare these results with the outcomes of investigations from permafrost regions (with more literatures) to observe the differences. 

Please give a little more Comparison on why the Conduction-Convection method might be better for this particular work. [You can also explain further with your results] 

The second objective could easily be divided into two (In line 95 and 96)

What are the implications of Amplitude and  smaller phase angles relative to the thermal diffusivity (for example in lines197 to 199)? … Could you be more explicit on the relationship between this parameters and the thermal diffusivity?

In the results section,  There is a need for the authors to be more explicit about the seasonal variation of the k particularly the phenomenon or phenomena responsible these variations differ for different locations.

The Day of the year DOY response that you Specifically mentioned to signify Certain periods of the year. I believe that the season of the year could also be emphasised upon such as to show the physical and possibly seasonal implications of the results for the benefits of the readers.

Minor Comments 

In Lines 18 and 43 starting the statement alphabet k appears to be awkward.

I would suggest that the statement should not start with single letter k. Please rephrase the statements  accordingly.

The abbreviation DOY which I would presume to be day of the year should be mentioned in full for the first time in the text before continuing with the abbreviation. 

The legends are not clearly showing what each line is showing. The plots Could be  generally made clearer (Fig 6 needs better improvement)

Reviewer 2 Report

Dear Authors,

The article entitled "Determination of long-term soil apparent thermal diffusivity  using near-surface soil temperature on the Tibetan Plateau", presents good results and contributes to the improvement of scientific knowledge regarding the knowledge of e of soil apparent thermal diffusivity.

However, I recommend that some improvements are made in order to improve the article.

minor corrections

Figure 4: Please improve the caption.

Figure 5: Please put the axes on the same scale.

Caption table 3: Please correct the superscript in the order of magnitude and in the unit of measure

Figure 6: The figure legend needs to be improved. Please insert Figure 6a, 6b, 6c etc... The figures in the first row can be on the same scale. The resolution, size, and quality too.

The captions of all figures are not good. I suggest making it more explicit and improving the resolution and size.

Page 4. Line 126: . The method used to determine soil apparent thermal diffusivity:

The authors must improve the details regarding obtaining parameters A1, A2 and φ.

General aspects that need more attention

Introduction

What are the main scientific contributions of the work? What's the innovation? I believe that the authors need to make clear the scientific gap that the article is filling in the literature.

Methodology

What is the primary soil information? Texture? Porosity? Residual moisture?

Were the moisture series shown calculated by the universal Topp equation?

Was there any correction or laboratory calibration of the moisture data?

I suggest inserting a flowchart or detailing the main methodological steps of the work.

Conclusion

Conclusions need to place more emphasis on the main results achieved. Therefore, I recommend it be more detailed.

Reviewer 3 Report

The thermal diffusivity is an important parameter for investigating soil surface heat transfer and temperature. This manuscript determined the thermal diffusivity (k) with a conduction-convection method using near-surface soil temperature on the Tibetan Plateau.

1.The equation (1) is a method considering both conduction and convection heat transfer, but this manuscript only determine the conduction thermal diffusivity (k), the convection heat transfer (W) was not be referred. Actually, there are many methods to determine the k, why chose this method to determine the k.

2.The soil thermal diffusivity was affected by soil water content and NDVI in the results section, in the discussion section, the soil water content was discussed, the NDVI was not be referred.

3.The manuscript has analyzed the relationship between in situ k and θ. In the TP plateau, when the low temperature under 0 , the water is the ice phase, how to get the water content with the soil water content sensor?

4.Table 1. the sensor depths for θ: 0/10 is 0.10

5.The symbol of in the equation2

Round 2

Reviewer 3 Report

The authors had revised the manuscript according to the comments.
The relationship between k and NDVI has been described in the Results, it is suggestion to discuss this results in the Discussion section.

Author Response

The authors had revised the manuscript according to the comments.
Response: We thank the reviewer for his/her comprehensive evaluation and thoughtful comments on our manuscript, which help tremendously to improve the quality of our work. We have tried our best to address reviewer’ concerns, one by one.

Besides, for clarity purpose, here we have listed reviewer’s comment in black, followed by our responses in blue, and the modifications to the manuscript are in italics.

The relationship between k and NDVI has been described in the Results, it is suggestion to discuss this results in the Discussion section.

Response: We moved the discussion for the relationship between k and NDVI to the Section 4.2, as follows,

“     Compared to BJ and NADORS, there were no apparent “outliers” at QOMS (Figure 6d-f)….. Thus, few “outliers” existed during the cold months at QOMS (Figure 6d-f).

       To indirectly investigate the effect of soil moisture on k from the vegetation aspect, the relationship between monthly k and NDVI was also examined (Figure 8), which is similar to the relationship between k to θ (Figure 7). The reason may be that NDVI is closely related to θ at these sites. The r’s between NDVI and θ ranged from 0.73 to 0.92 on monthly timescale, and from 0.69 to 0.82 on annual timescale (Table 4). Compared to the other two sites, more vegetation was present in BJ, so the NDVI of BJ represented more vegetation information than bare soil. Therefore, the correlation between k and NDVI was weaker than that between k and θ at BJ, while closer to that between k and θ at the other two sites, as shown in Figures 7-8.

      With in situ soil temperature, we determined k by the conduction-convection method for 2014-2016….”

The two paragraphs in the Result Section (“Interestingly, the monthly k had similar correlation with NDVI as it did with θ. The reason may be that NDVI was closely related to θ at these sites. The r’s between NDVI and θ ranged from 0.73 to 0.92 on monthly a timescale, and from 0.69 to 0.82 on annual timescale, as shown in Table 4.

      At QOMS and NADORS, the r’s of k vs NDVI were close to those of k vs θ (∆r < 0.08), while r’s of k vs NDVI were smaller than those of k vs θ at BJ. Compared to the other two sites, more vegetation was present in BJ, so the NDVI at BJ represented more vegetation information than bare soil. Therefore, the correlation between k and NDVI was weaker than that between k and θ at BJ.”) were revised as follows,

“       Interestingly, the monthly k had similar correlation with NDVI as it did with θ. At QOMS and NADORS, the r’s of k vs NDVI were close to those of k vs θ (r < 0.08), while r’s of k vs NDVI were smaller than those of k vs θ at BJ.